# *Zucchini Yellow Mosaic Virus* (ZYMV) as a Serious Biotic Stress to Cucurbits: Prevalence, Diversity, and Its Implications for Crop Sustainability

**DOI:** 10.3390/plants12193503

**Published:** 2023-10-08

**Authors:** Muhammad Ahsan, Muhammad Ashfaq, Mahmoud Ahmed Amer, Muhammad Taimoor Shakeel, Mirza Abid Mehmood, Muhammad Umar, Mohammed Ali Al-Saleh

**Affiliations:** 1Institute of Environmental and Agricultural Sciences, University of Okara, Okara 56300, Pakistan; ahsan.kashi@gmail.com; 2Department of Plant Pathology, Balochistan Agriculture College, Quetta 87100, Pakistan; 3Plant Pathology, Institute of Plant Protection, Muhammad Nawaz Shareef University of Agriculture, Multan 61000, Pakistan; abid.mehmood@mnsuam.edu.pk; 4Plant Protection Department, College of Food and Agriculture Sciences, King Saud University, P.O. Box 2460, Riyadh 11451, Saudi Arabia; ruamerm@ksu.edu.sa (M.A.A.); malsaleh@ksu.edu.sa (M.A.A.-S.); 5Department of Plant Pathology, Faculty of Agriculture and Environment, The Islamia University of Bahawalpur, Bahawalpur 63100, Pakistan; taimoor.shakeel@iub.edu.pk; 6Biosecurity Tasmania, Department of Natural Resources and Environment, Hobart, TAS 7008, Australia; m.umar@utas.edu.au

**Keywords:** RNA, ZYMV, coat protein, in silico restriction, subgroup IIa and IIb, cucurbits, biotic stress

## Abstract

*Zucchini yellow mosaic virus* (ZYMV) is a severe threat to cucurbit crops worldwide, including Pakistan. This study was pursued to evaluate the prevalence, geographic distribution, and molecular diversity of ZYMV isolates infecting cucurbits in Pakistan’s Pothwar region. Almost all the plant viruses act as a biotic stress on the host plants, which results in a yield loss. These viruses cause losses in single-infection or in mixed-infection cucurbit crops, and we have found a number of mixed-infected samples belonging to the *Curubitaceae* family. Serological detection of the tested potyviruses in the collected cucurbit samples revealed that ZYMV was the most prevalent virus, with a disease incidence (DI) at 35.2%, followed by *Papaya ringspot virus* (PRSV) with an incidence of 2.2%, and *Watermelon mosaic virus* (WMV) having an incidence as little as 0.5% in 2016. In the year 2017, a relatively higher disease incidence of 39.7%, 2.4%, and 0.3% for ZYMV, WMV, and PRSV, respectively, was recorded. ZYMV was the most prevalent virus with the highest incidence in Attock, Rawalpindi, and Islamabad, while PRSV was observed to be the highest in Islamabad and Jhelum. WMV infection was observed only in Rawalpindi and Chakwal. Newly detected Pakistani ZYMV isolates shared 95.8–97.0% nucleotide identities among themselves and 77.1–97.8% with other isolates retrieved from GenBank. Phylogenetic relationships obtained using different ZYMV isolates retrieved from GenBank and validated by in silico restriction analysis revealed that four Pakistani isolates clustered with other ZYMV isolates in group IIb with Chinese, Italian, Polish, and French isolates, while another isolate (MK848239) formed a separate minor clade within IIb. The isolate MK8482490, reported to infect bitter gourd in Pakistan, shared a minor clade with a Chinese isolate (KX884570). Recombination analysis revealed that the recently found ZYMV isolate (MK848239) is most likely a recombinant of Pakistani (MK848237) and Italian (MK956829) isolates, with a recombinant breakpoint between 266 and 814 nucleotide positions. Local isolate comparison and recombination detection may aid in the development of a breeding program that identifies resistant sources against recombinant isolates because the ZYMV is prevalent in a few cucurbit species grown in the surveyed areas and causes heavy losses and economic damage to the agricultural community.

## 1. Introduction

Vegetables are high-value crops that frequently provide sustainable livelihoods for smallholders and large-scale commercial farmers. Cucurbits (*Cucurbitaceae* family) are major vegetable and fruit crops in Pakistan, grown throughout the country, including the Pothwar region [1]. This family includes around 95 extant genera with over 900 species [2]. The Pothwar region’s favorable climatic conditions facilitate the production of most cucurbits, cultivated as summer and winter crops throughout the year [3,4]. The main cultivated cucurbit species of the Pothwar region include melon (*Cucumis melo* L.), cucumber (*C. sativus* L.), squash (*Cucurbita* sp.), gourd (*Luffa* sp.), pumpkin (*Cucurbita moschata*), round gourd (*Praecitrullus fistulosus* (Stocks) Pangalo) and watermelon (*Citrullus lanatus* L.). Cucurbits play a vital role in the diet, as they contain vitamins and other dietary constituents (vitamins B and C, niacin, protein, lipid, carbohydrate, Ca, Fe, P) [5]. Several biotic and abiotic factors may limit cucurbit production. Among biotic factors, earlier studies reported approximately 90 different viruses infecting cucurbits from diverse regions of the world [4,6,7]. The cucurbit average yield in Pakistan is relatively lower than in other countries due to the region’s prevalence of diseases by RNA viruses. Among these RNA viruses, the most common are *Zucchini yellow mosaic virus* (ZYMV; Potyvirus), *Cucurbit aphid-borne yellows virus* (CABYV; Polerovirus), *Papaya ringspot virus* (PRSV; Potyvirus), *Cucumber mosaic virus* (CMV; Cucumovirus), *Cucurbit* chlorotic yellows virus (CCYV), and *Watermelon mosaic virus* (WMV; Potyvirus) [7,8,9,10,11,12].

ZYMV, belonging to genus *Potyvirus* in the family *Potyviridae*, is reported to incite severe diseases in crops grown as tropical and subtropical cucurbits, globally [13,14,15]. Virus particles are flexuous filaments of sizes ranging from 680 to 730 nm in length [13,16]. The potyviruses are reported to have been transmitted through numerous aphid species non-persistently and by seeds [17,18,19,20]. The aphid vectors play their role in spreading the virus within the cultivated fields and from cucurbit crops to weeds, acting as natural reservoirs of the virus during the offseason, and work as efficient sources of infection at the start of the growing season [14,21,22]. ZYMV causes typical symptoms of mosaic, leaf distortion, blisters, chlorosis on leaves, and general stunting in the affected cucurbit plants. In severe systemic infection, affected fruits develop malformation and surface discoloration, making them unmarketable [7,14,23]. Yield losses caused by ZYMV can range from mild to severe, with some reports suggesting reductions of up to 80% in affected crops [24]. The magnitude of yield losses depends on several factors, including the prevalence and virulence of the virus, susceptibility of cultivated varieties, agronomic practices, and environmental conditions [25]. Though a few earlier studies revealed the presence of ZYMV infection in bottle gourd [1], melon [26], ridge gourd [10], and round gourd [27] in other parts of the country, there is a critical need for an intensive study highlighting the comprehensive understanding of ZYMV prevalence in major cucurbits to devise proper management strategies. So, this study was conducted with the aim of providing an estimation of the incidence and molecular characterization of potyviruses, especially ZYMV, in the major cucurbit crops of the Pothwar region.

## 2. Results

### 2.1. Incidence and Distribution of Potyviruses

The result of general serological screening done by PTA–ELISA showed that, out of the 767 leaf samples collected from cucurbit plants exhibiting possible symptoms of viral diseases, 290 samples (37.8%) were infected with potyviruses in 2016. These positive samples were retested with specific DAS–ELISA and the incidence of ZYMV, PRSV, and WMV in the collected cucurbit samples was recorded to be 35.2%, 2.2%, and 0.5%, respectively (Table 1). In the year 2017, out of 795 samples, ZYMV was the most prevalent virus, with a relatively increased disease incidence of 39.7%, followed by WMV (2.4%) and PRSV (0.3%) (Table 1). In the Attock district, smooth gourd was found to be highly infected with potyviruses, followed by ridge gourd, round gourd, and cucumber, while bitter gourd was the lowest infected crop during 2016 and 2017. ZYMV was the most prevalent virus found heavily infecting ridge gourd, followed by smooth gourd, round gourd, and cucumber, while melon was the less-infecting crop in the Chakwal district during both sampling years. In the Jhelum and Islamabad districts, melon was observed without any potyvirus infection (Table 1).

The most common disease was ZYMV, which had the highest incidence in Attock, followed by Rawalpindi, Islamabad, and Chakwal, and the lowest incidence in Jhelum. The highest incidence of PRSV was detected in Islamabad, followed by Jhelum, Attock, and Chakwal, while in Rawalpindi, cucurbit crops were less infected with PRSV. WMV infection was observed only in Rawalpindi and Chakwal (Table 1).

### 2.2. Molecular Characterization of Zucchini Yellow Mosaic Virus (ZYMV)

PCR results revealed that ZYMV-specific primers [1] amplified a specific fragment of a size of 1.5 kb in all the ELISA-positive samples chosen from each surveyed location. Five sequences of ZYMV were acquired when amplified fragments were processed by Sanger sequencing in both directions. Five sequences of ZYMV were sequenced and subjected to BLASTn to confirm that all sequenced samples containing partial NIb, 3′UTR genomic regions, and complete CP showed nucleotide identities of 94.5–99.9% to the corresponding regions of other ZYMV isolates (same subgroup) from GenBank. After careful analysis, five ZYMV present-study isolates were submitted to GenBank (Table 2). Each sequence of ZYMV comprised of 211 nt of 3′UTR, 837 nt encoding 278 aa of CP, 363 nt encoding 121 aa of NIb, and followed by a poly-A tail of varying length. The proportions of guanine, uracil, adenine, and cytosine in the CP gene sequences of these isolates were 25.5%, 22.5%, 32.4%, and 19.5%, respectively (Table 2).

Moreover, the sequence identity matrix study showed that all the Pakistani ZYMV isolates shared a 95.8 to 97.0% similarity. Three recently identified isolates (MK848237, 97.6%; MK848238, 97.8%; MK848241, 97%) showed the highest similarity with the Pakistani ZYMV isolate AIRGPK (KR261952) reported in the year 2017 from round gourd, while MK848239 shared the highest similarity (96.7%) with the KX884565 isolate, reported to infect crayfish from China, and MK848240 shared maximum similarity (97.3%) with the French, Iranian, and South Korean isolates. All the newly characterized isolates showed the lowest similarity with sequence JF797206 isolated from *cucumber* in India (Table 3, Figure 1). We observed a 98.9–99.4% identity in the amino acid sequences of the CP genes of all isolates, while the formerly reported subgroup IIb ZYMV isolates shared 93.9–99.6% identities. Amino acid sequence identities of 88.1–92.4% and 88.5% were observed with the subgroup IIa and group I isolates, respectively, from other regions of the world.

### 2.3. Phylogenetic Analysis and In Silico RFLP-Based Phylogeny-Based Comparison

Phylogenetic analysis revealed that all the sequences were outlined into two major groups, i.e., group I and group II. Group I consisted of a single isolate reported from China, while group IIa included 18 isolates reported from Japan, the United Kingdom, Australia, the United States, Iran, Egypt, Mali, Italy, Israel, India, Canada, Turkey, and France. Group IIb also contained 18 isolates, including 5 new and 1 previously reported from Pakistan, which were scattered among several minor clades. One of the isolates (MK848239), reported to infect pumpkin in Pakistan, was found to be relatively divergent, as it did not share its position in a minor clade within group IIb with any other isolate. Another isolate (MK8482490), reported to infect bitter gourd in Pakistan, shared a minor clade with a Chinese isolate (KX884570). Four other newly reported isolates from Pakistan infecting watermelon, cucumber, smooth gourd, and round gourd shared a separate minor clade with isolates of ZYMV reported from Poland and France.

In silico RFLP simulation of ZYMV nucleotide sequences showed the presence of seven to nine cut positions of each enzyme. Virtual gel resulting from RFLP simulation revealed that nearly all the ZYMV isolates contained a distinct 43 bp band, while the CMV outgroup isolates lacked it. Similarly, group I exhibited a unique 99 bp fragment, whereas a 48 bp fragment distinguished group IIa isolates from the others (Figure 2). Phylograms (Figure 3) obtained after the virtual gel analysis in PyElph v1.4 affirmed the findings of the phylogeny analysis, i.e., the clustering of isolates in two main clades, except the shuffling of group IIa isolates to IIb isolates in minor clades. The AY995216 isolate from New Zealand also shifted from IIb to group I, clustering with the Chinese Vege isolate. This shift of isolate clustering may be due to silent mutation in the sequences of these isolates at restriction points [28]. Amino acid analysis confirms the occurrence of specific potyvirus motifs: QMKAAA in all the Pakistani ZYMV isolates, the CP region’s MVWCIEN, and the NIb area’s GNNS are the main criteria for potyvirus identification [29,30].

### 2.4. Recombination Analysis

Recombination analysis showed that our isolate AAAP (MK848239) is likely to be recombinant with Pakistani (MK848237) and Italian (MK956829) ZYMV pumpkin isolates (Figure 4). Five statistical methods, i.e., Chimaera, GENECONV, MaxChi, 3SEQ, and SiScan, confirmed this recombination event, with a recombination breakpoint between nucleotides 266 and 814. A second recombination event was also detected between nucleotide 74–650 through four statistical methods in RDP4, which showed that the Pakistani KR261952 isolate reported in our previous study [27] is likely a recombinant, originating from the Australian (MN598576) and Pakistani (MK848239) ZYMV pumpkin isolate (Table 4).

## 3. Discussion

As with all viruses, plant viruses implement their mechanism of highjacking the nuclear machinery of the host cells and dedicate the plant resources for the replication and movement of its particles to new cells and tissues for the sake of invasion. Apart from the direct effects, a number of genes present in potyviruses are reported to interact with Ca^2+^ sensors and activate the related transcriptional procedures. Few genes which show altered transcribed abundance are related to the process of photosynthesis, signaling, and defense responses [31]. As a consequence of potyvirus infection, the photosynthetic and defense machinery is under stress, leading to yield loss in the longer run. Among plant viruses, the genus Potyvirus is considered one of the largest groups of plant viruses, with a large number of definite and tentative candidates. Among the members of potyviruses, ZYMV, WMV, and PRSV have been reported to be devastating to several important cucurbit crops, particularly from the economic point of view [32,33]. ZYMV may cause up to 80% yield losses in different crops, depending on the crop germplasm, age, viral load, vector population, and favorable environmental conditions [24,25,34]. The classification of potyviruses has sometimes led to confusion because of their striking variations in symptomatology, host range, and genetic makeup, with markable changes among the strains and the tentative candidates assigned to this genus [35]. Considering the diverse nature of this genus, the use of molecular approaches has been a successful tool for characterizing the local potyvirus isolates infecting cucurbits. This study helped in estimating the status of the prevalence of three species of potyviruses infecting cucurbits grown in the Pothwar region using serological identification and molecular characterization, which aided in understanding the evolutionary and recombination histories of the identified isolates in recent years. 

The results of surveys in the Pothwar region revealed that ZYMV was the most prevalent virus, with a disease incidence of 37.52%, followed by PRSV (2.3%) and WMV (0.38%) in 2016–17, which showed that ZYMV is the most widespread among the tested viruses in the Pothwar region infecting cucurbits, and in line with the reports of other scientists working on the identification of this virus in the country [1,10,27,34]. Considering the disease-incidence data of the potyviruses, all crops other than cucumber have a substantially lower proportion of disease, implying that cucumber crop cultivation is vulnerable to these diseases, although other cucurbit crops may hold a desirable position in the region. Sequence identity matrix findings indicate that Pakistani ZYMV isolates exhibited 94.6–97.9% identities with other ZYMV isolates retrieved from GenBank, while they shared 95.8–97.0% nucleotide identities with one another. Interestingly, the similarity data reveal that one of the closely related isolates was reported as early as 1990 from the USA, which had a 95.8% (on average) similarity with all the newly reported isolates from Pakistan. It can be concluded that the virus has a good level of adaptation to several cucurbit hosts cultivated in different countries over the last three decades; however, the similarity percentage did not go below 94.6%. It has also been reported that variation in the genomic regions, viz. the CP gene, helper component–protease (HC-Pro), or P3 protein genes of plant viruses, especially ZYMV, leads to varying expression of symptoms, transmission, and host range [14,23].

In the nucleotide-based phylogenetic analysis, it was revealed that the five new Pakistani isolates of ZYMV, along with thirty-two other isolates retrieved from GenBank, NCBI, produced two distinct groups, including group I and group II. Group I includes a single isolate reported from China, while group II has two subgroups, group IIa and b. Group IIa comprises of 18 isolates reported from several countries and which are relatively different from the isolates reported from Pakistan, regardless of the host. Whereas, in group IIb, all isolates were reported to infect cucumber, watermelon, pumpkin, bitter gourd, round gourd, and smooth gourd Pakistan. It was observed that the isolates reported from Pakistan are phylogenetically related to each other, and apparently the host has no impact on the development of new strains. The phylogenetic tree depicts a low genetic diversity among the isolates of ZYMV reported to infect several hosts from different countries. The overall picture of the phylogenetic tree shows that isolates of ZYMV characterized from several hosts in different countries over the years are different, but the level of variance is not very high. These phylogeny results do not highlight any fact about the route of the virus entry into the country, because almost all the isolates share similar levels of identity. 

The clustering pattern of isolates were validated using the in silico RFLP simulation of the obtained sequences, which revealed the similar pattern, except for the isolates of the IIa group, most of which clustered in group IIb, along with divergent patterns of isolate AY995216, which clustered with group I from IIb. This shift may be due to a silent mutation in the sequences of these isolates at restriction points. Silent mutations can occur at restriction-enzyme-recognition sequences, allowing the introduction or removal of restriction sites without affecting the protein translation [28]. Integration of information from phylogeny and virtual restriction simulations helped researchers to identify clustering patterns and variations, including the clustering of isolates in different groups and the impact of silent mutations on the clustering pattern [28,36].

Maina, et al. [37,38] recorded recombination events in the CP, as well as in the NIb, CI, P1, and P3 regions of the ZYMV isolates, and they also confirmed the formation of new groups due to these recombinant isolates. Variation may appear in the genome due to the mutation, recombination, and reassortment of viral genome elements [39]. The quarantine department must take serious testing and legislation measures to control the invasion of new viruses to the country.

Recombination analysis aids in the study of host adaptation and virus evolution in the abolition or emergence of variation in the viral genome. In this work, CP-gene-based recombination analysis revealed the detection of two recombination events, indicating that ZYMV pumpkin (MK848239) and ZYMV round gourd (KR261952) isolates are likely to be recombinant. Similar results of recombination in CP gene sequences have been elaborated in other potyviruses, i.e., PRSV [40,41], *Plum pox virus* (PPV) [42], Sugarcane mosaic *virus* (SCMV) [43,44], *Yam mosaic virus* (YMV) [45], *Bean yellow mosaic virus* (BYMV) [46], *Potato virus Y* (PVY) [47], and *Chilli veinal mottle virus* (ChiVMV) [48]. The fitness of potyvirus populations in changing conditions confirmed the importance of recombination studies [49]. The presence of recombinant isolates and various groups of ZYMV isolates with widespread distribution creates an alarming situation for productive crop yields that must be remedied through suitable viral detection and management strategies. Moreover, the quarantine department must implement rigorous testing and regulatory measures to prevent the introduction of new viruses into the country.

## 4. Materials and Methods

### 4.1. Surveys and Collection of Important Vegetable Crop Samples

Vegetable-growing areas of the Pothwar region (Figure 5) were surveyed using a random stratified design during 2016–2017 [50] to record the incidence and distribution of potyviruses, as per the scheme given in Figure 6.

A total of 767 and 795 leaf samples were collected in years 2016 and 2017, respectively, from eight to ten randomly selected vegetable fields in five districts of Pothwar region of Punjab, including Islamabad, Rawalpindi, Chakwal, Attock, and Jhelum (Table 1). The samples were collected based on typical virus-like symptoms, such as ring spots, mosaic, chlorotic, mottling, stunting, and puckering. All the collected samples were stored in iceboxes and brought to the Plant Virology lab, Department of Plant Pathology, PMAS Arid Agriculture University, Rawalpindi, Pakistan, for further processing. The samples were kept at −20 °C until they were processed for viral agent detection and characterization.

### 4.2. Serological Diagnosis of RNA Viruses

All the collected plant samples were initially tested serologically using a group-specific ELISA kit, i.e., potyvirus group ELISA kit (art. no. 163072, Bioreba, Reinach, Switzerland), and later by virus-specific ELISA using commercially available kits for ZYMV (art. no. 161272, Bioreba), PRSV (art. no. 151972, Bioreba), and WMV (art. no. 161172, Bioreba), with slight modification in the method of Clark and Adams [51]. The optical density (OD 405 nm) of the yellow color was determined using the ELISA reader with positive and negative controls (Bioreba, Switzerland) as a reference [52]. The samples with a 2X or greater OD 405 nm value than the healthy (control) samples were referred to as positive. ELISA-positive samples were used to determine the disease incidence percentage using the proportionate formula as given by Ahsan et al. [53]:(1)$$\mathrm{Disease}\;\mathrm{incidence}\;\left(\%\right)=\frac{\mathrm{Number}\;\mathrm{of}\;\mathrm{plants}\;\mathrm{infected}}{\mathrm{Total}\;\mathrm{number}\;\mathrm{of}\;\mathrm{plants}\;\mathrm{tested}}\;\times \;100$$

### 4.3. RNA Extraction and Molecular Characterization of ZYMV

After the serological screening, five representative samples infected with ZYMV were selected for molecular characterization based on coat protein (CP) gene sequences to ascertain genetic diversity. For this purpose, total RNA was extracted from ELISA-positive ZYMV samples by using TRIzol Reagent (Thermo Scientific, Waltham, MA, USA) according to the manufacturer’s protocol. First-strand complementary DNA (cDNA) was synthesized using the RevertAid RT Reverse Transcription Kit (Thermo Scientific, USA) and ZYMV CP downstream primer (ZYMV R Not 1-25, 5′- AAC TGG AAG AAT TCG CGG CCG CAC GAAT) [1] by following instructions of the manufacturer. The CP gene was amplified by the primer pair (ZYMV F-8193-22; 5′-CACCAAGCAATGYTRGTTG-3′ and ZYMV R Not 1-25) [1] in a thermal cycler using GreenTaq Green PCR Master Mix (2X) (Thermo Scientific, USA) following the reaction conditions; 94 °C for 3 min followed by 35 cycles of 94 °C for 30 s, 52 °C for 45 s, and 72 °C for 90 s, and a final extension at 72 °C for 10 min. RT-PCR products were run through 1% agarose gel added with ethidium bromide (100 μg mL^−1^). The amplicons of ~1.50 kb was purified using the GeneJET PCR Purification Kit (Thermo Scientific, USA) and cloned in the pTZ57R/T vector (InsTAclone^TM^ PCR cloning kit, Thermo Scientific, USA) with *E. coli* strain XL1-Blue competent cells. The manufacturer’s instructions were followed to isolate the recombinant plasmid DNA using the GeneJET Plasmid Miniprep Kit (Thermo Scientific, USA). ECoR1 and HindIII enzymes were used for the restriction digestion and validation of transformants, while M13 forward and reverse primers were employed for the sequencing of positive clones in both directions from Macrogen (Seoul, Republic of Korea).

The ExPASy translate tool [54] was employed to obtained amino acid sequences from nucleotides and subjected to BLASTn [55] and BLASTp [56] analysis for identification and to calculate sequence identity percentages. A total of 32 resembling ZYMV CP gene sequences from the NCBI database were downloaded and aligned with 6 isolates that were newly identified from Pakistan, along with an isolate of CMV (outgroup) using the CLUSTAL W program embedded in the MEGA7 software [57]. The aligned sequences along with a CMV isolate as an outgroup were subjected to phylogenetic reconstructions using the Maximum Likelihood with the Jukes–Cantor model, supported by robustness assessments, through 1000 bootstrap replicates. The identities of nucleotide and amino acid sequences were determined with the Sequence Identity Matrix option in BioEdit 7.2 [58].

### 4.4. In Silico RFLP and Recombination Analysis

The CP-based in silico restriction fragment length polymorphism (RFLP) of new Pakistani isolates and 32 retrieved sequences of ZYMV isolates from GenBank were executed using the DdeI, HinfI, and HpaII restriction enzymes in CLC Main Workbench 20 (QIAGEN, Aarhus, Denmark). The resulting gel (Figure 3) showed the presence of 3-3-5 bands of different sizes for each isolate analyzed in PyElph v1.4 [36] for the construction of phylograms and serotyping of isolates. Recombinant events in the 5 new Pakistani and 32 other ZYMV isolates were analyzed with RDP4 [59] with default settings, viz. RDP, GENECONV, BootScan, MaxChi, Siscan, Chimaera, and 3SEQ.

## 5. Conclusions

ZYMV is the most prevalent virus infecting all the major cucurbits grown in the Pothwar region of Pakistan. In silico analysis of ZYMV sequences revealed the presence of five new group A and B isolates closely related to isolates from Pakistan, China, Japan Poland, Italy, and Australia. Recombinant strain detection and higher incidence of the virus across the Pothwar region underscore the disease’s magnitude, forcing us to formulate new breeding programs to identify resistant sources and apprise our farmers about the disease.

## Figures and Tables

**Figure 1 plants-12-03503-f001:**
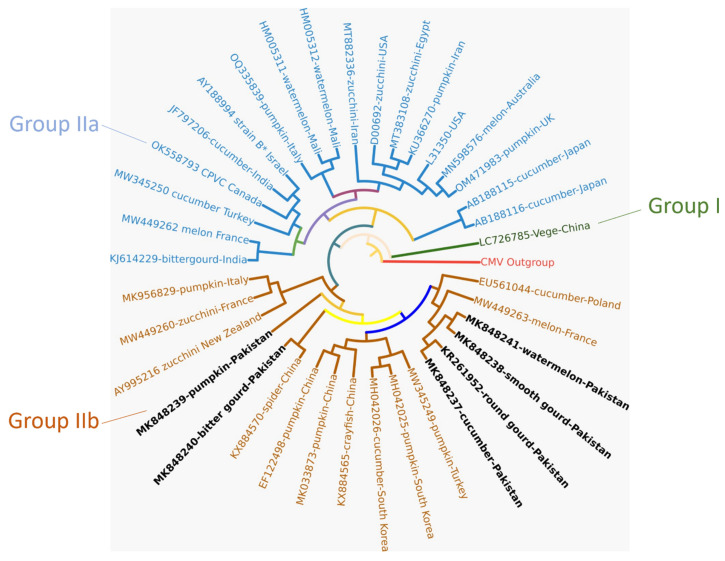
The phylogenetic analysis of 37 isolates of ZYMV using the maximum-likelihood method with the Jukes–Cantor model, supported by robustness assessments, keeping the bootstrap value at 1000.

**Figure 2 plants-12-03503-f002:**
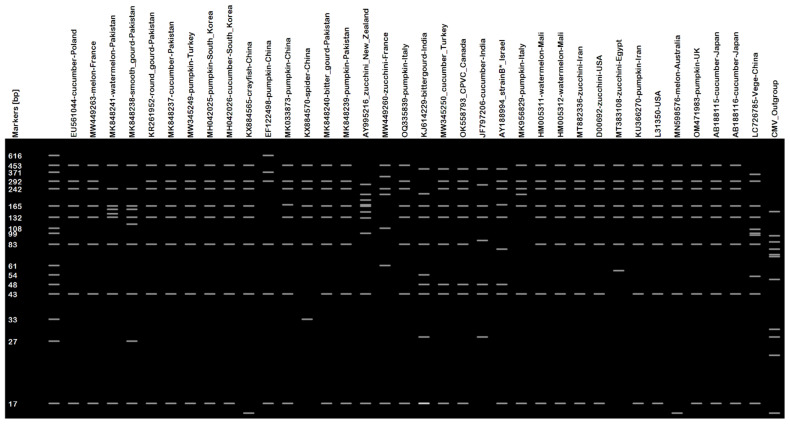
Virtual gel after the in silico RFLP simulation of CP genes of the ZYMV isolates.

**Figure 3 plants-12-03503-f003:**
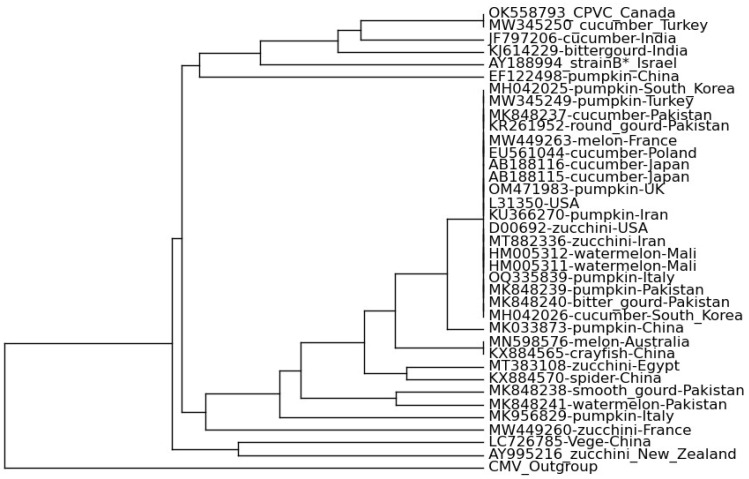
In silico RFLP simulation and genotyping of 37 isolates of ZYMV based on the CP gene.

**Figure 4 plants-12-03503-f004:**
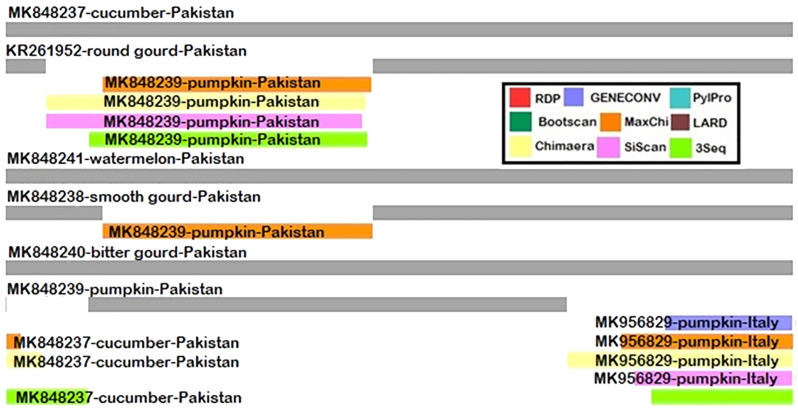
Recombinant events detected in ZYMV isolates using Standard Methods in RDP4.

**Figure 5 plants-12-03503-f005:**
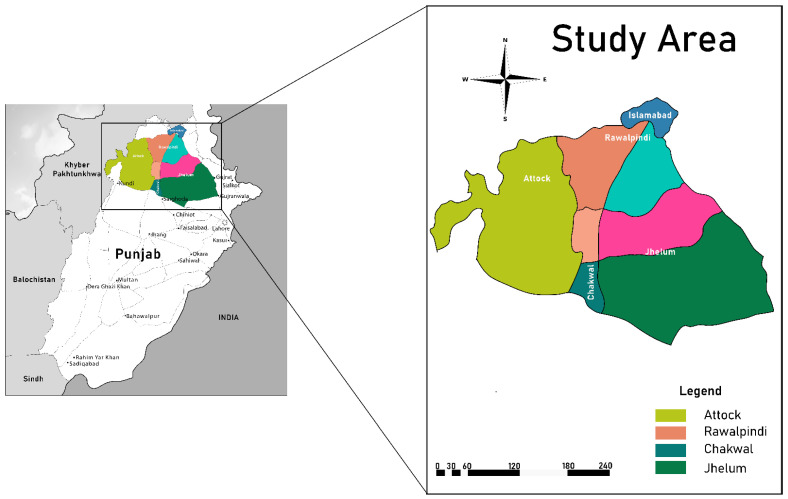
Map of the Pothwar region (colored) in the Punjab province of Pakistan.

**Figure 6 plants-12-03503-f006:**
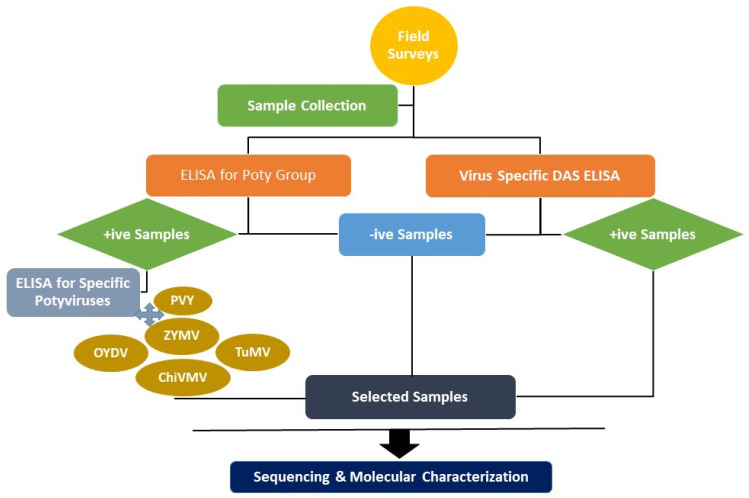
Scheme used to detect plant RNA viruses that infect vegetable samples from the region of Pothwar in Pakistan.

**Table 1 plants-12-03503-t001:** Disease incidence of potyviruses infecting cucurbits in 2016–2017 from the Pothwar region of Pakistan.

Location	Crop	D.I. % in 2016					D.I. % in 2017				
		Potyvirus ^a^		ZYMV ^b^	WMV ^c^	PRSV ^d^	Potyvirus		ZYMV	WMV	PRSV
		Samples *	D.I. %				Samples *	D.I. %			
Attock	Cucumber	18 (40)	45.0	40.0	0.0	5.0	17 (41)	41.5	41.5	0.0	2.4
	Round gourd	17 (32)	53.1	50.0	0.0	3.13	18 (33)	54.5	54.5	0.0	0.0
	Watermelon	8 (21)	38.1	28.6	0.0	9.5	9 (21)	42.9	33.3	0.0	9.5
	Melon	6 (21)	28.6	19.0	0.0	9.5	8 (22)	36.3	22.7	0.0	13.6
	Pumpkin	3 (10)	30.0	30.0	0.0	0.0	5 (13)	38.5	38.5	0.0	0.0
	Bitter gourd	2 (20)	10.0	10.0	0.0	0.0	3 (21)	14.3	14.3	0.0	0.0
	Ridge gourd	30 (38)	78.9	78.9	0.0	0.0	32 (39)	82.1	82.1	0.0	0.0
	Smooth gourd	5 (6)	83.3	83.3	0.0	0.0	7 (8)	87.5	87.5	0.0	0.0
Chakwal	Cucumber	6 (25)	24.0	24.0	0.0	0.0	7 (22)	31.8	31.8	0.0	0.0
	Round gourd	6 (17)	35.3	35.3	0.0	0.0	4 (16)	25.0	25.0	0.0	0.0
	Watermelon	4 (19)	21.1	21.1	5.3	0.0	5 (18)	27.8	27.8	0.0	5.6
	Melon	1 (11)	9.1	0.0	0.0	9.1	2 (12)	16.7	8.3	0.0	8.3
	Pumpkin	2 (13)	15.4	15.4	0.0	0.0	3 (13)	23.1	23.1	0.0	0.0
	Bitter gourd	1 (16)	6.3	6.3	0.0	0.0	2 (16)	12.5	12.5	0.0	0.0
	Ridge gourd	15 (25)	60.0	60.0	0.0	0.0	19 (26)	73.1	73.1	0.0	0.0
	Smooth gourd	4 (10)	40.0	40.0	0.0	0.0	5 (8)	62.5	62.5	0.0	0.0
Jhelum	Cucumber	5 (21)	23.8	23.8	0.0	0.0	6 (25)	24.0	24.0	0.0	0.0
	Round gourd	3 (7)	42.9	42.9	0.0	0.0	4 (9)	44.4	44.4	0.0	0.0
	Watermelon	4 (21)	19.0	0.0	0.0	19.0	5 (21)	23.8	9.5	0.0	14.3
	Melon	0 (5)	0.0	0.0	0.0	0.0	0 (8)	0.0	0.0	0.0	0.0
	Pumpkin	1 (14)	7.1	7.1	0.0	0.0	2 (15)	13.3	13.3	0.0	0.0
	Bitter gourd	1 (16)	6.3	6.3	0.0	0.0	2 (16)	6.3	6.3	0.0	0.0
	Ridge gourd	14 (26)	53.8	53.8	0.0	0.0	15 (25)	60.0	60.0	0.0	0.0
	Smooth gourd	3 (9)	33.3	33.3	0.0	0.0	3 (9)	33.3	33.3	0.0	0.0
Rawalpindi	Cucumber	23 (68)	33.8	32.4	1.5	0.0	25 (69)	36.2	36.2	1.4	1.4
	Round gourd	13 (26)	50.0	50.0	0.0	0.0	15 (29)	51.7	51.7	0.0	0.0
	Watermelon	11 (29)	37.9	27.6	3.4	6.9	12 (30)	40.0	30.0	3.3	10.0
	Melon	3 (18)	16.7	5.6	5.6	5.6	4 (21)	19.0	9.5	0.0	9.5
	Pumpkin	3 (15)	20.0	20.0	0.0	0.0	4 (14)	28.6	28.6	0.0	0.0
	Bitter gourd	3 (28)	10.7	10.7	0.0	0.0	4 (29)	13.8	13.8	0.0	0.0
	Ridge gourd	30 (37)	81.1	81.1	0.0	0.0	33 (38)	86.8	86.8	0.0	0.0
	Smooth gourd	8 (12)	66.7	66.7	0.0	0.0	10 (15)	66.7	66.7	0.0	0.0
Islamabad	Cucumber	5 (13)	38.5	38.5	0.0	0.0	2 (11)	18.2	18.2	0.0	0.0
	Round gourd	4 (9)	44.4	44.4	0.0	0.0	6 (12)	50.0	50.0	0.0	0.0
	Watermelon	2 (8)	25.0	0.0	0.0	25.0	3 (8)	37.5	12.5	0.0	25.0
	Melon	0 (7)	0.0	0.0	0.0	0.0	0 (6)	0.0	0.0	0.0	0.0
	Pumpkin	2 (7)	28.6	28.6	0.0	0.0	3 (8)	37.5	37.5	0.0	0.0
	Bitter gourd	2 (15)	13.3	13.3	0.0	0.0	3 (15)	20.0	20.0	0.0	0.0
	Ridge gourd	15 (23)	65.2	65.2	0.0	0.0	18 (24)	75.0	75.0	0.0	0.0
	Smooth gourd	7 (9)	77.8	77.8	0.0	0.0	7 (9)	77.8	77.8	0.0	0.0
Total % D. I.		290 (767)	37.8	35.2	0.5	2.2	331 (795)	41.6	39.7	0.3	2.4

* Samples: no. of +ive samples (Total tested samples); ^a^ Disease incidence of Potyvirus = (no. of Potyvirus +ive samples/Total tested samples) × 100; ^b^ Disease incidence of ZYMV = (no. of ZYMV +ive samples/Total tested samples) × 100; ^c^ Disease incidence of WMV = (no. of WMV +ive samples/Total tested samples) × 100; ^d^ Disease incidence of PRSV = (no. of PRSV +ive samples/Total tested samples) × 100.

**Table 2 plants-12-03503-t002:** Features of Pakistani ZYMV isolates submitted in GenBank.

Accession No.	Isolate	Host	Viral cDNA (nt)	NIb		Coat Protein Gene						3′UTR (nt)
				Nt	aa	CP (nt)	CP (aa)	U (%)	C (%)	A (%)	G (%)	
MK848237	AARC	Cucumber	1426	363	121	837	278	23.42	18.64	32.97	24.97	226
MK848238	AAHSG	Smooth Gourd	1426	363	121	837	278	23.42	19.12	32.74	24.73	226
MK848239	AAAP	Pumpkin	1426	363	121	837	278	23.54	19.12	32.26	25.09	226
MK848240	AARBG	Bitter Gourd	1426	363	121	837	278	23.78	18.76	31.78	25.69	226
MK848241	AAHWM	Watermelon	1426	363	121	837	278	23.54	19.0	32.50	24.97	226

**Table 3 plants-12-03503-t003:** Nucleotide identity of new Pakistani ZYMV isolates with isolates reported to infect several hosts in different countries around the world.

Accession No.	Location	Host	Year	NT Identity % Age				
				MK848237	MK848238	MK848239	MK848240	MK848241
KR261952	Pakistan	Round Gourd	2014	97.0	96.8	96.1	95.2	97.0
MK956829	Italy	Pumpkin	2019	97.6	97.8	96.1	97.2	97.6
HM005312	Mali	Watermelon	2009	94.7	94.8	94.5	95.2	94.7
HM005311	Mali	Watermelon	2009	95.8	95.5	95.3	96.8	95.8
EU561044	Poland	Cucumber	2008	95.5	94.9	94.9	96.8	95.5
EF122498	China	Pumpkin	2006	96.5	96.8	96.5	96.7	96.5
MW345249	Turkey	Pumpkin	2011	97.2	96.8	96.5	96.9	97.2
MH042025	Republic of Korea	Pumpkin	2016	96.9	97.0	96.3	97.2	96.9
MH042026	Republic of Korea	Cucumber	2016	97.1	97.0	96.5	97.3	97.1
KX884565	China	Crayfish	2014	97.0	96.9	96.5	97.3	97.0
KX884570	China	Spider	2013	95.6	95.6	95.0	95.6	95.6
AB188116	Japan	Cucumber	2004	97.3	97.0	96.7	97.2	97.3
AB188115	Japan	Cucumber	2004	96.1	95.4	95.5	96.5	96.1
MT383108	Egypt	Zucchini	2018	96.1	95.4	95.6	96.5	96.1
MN598576	Australia	Melon	2019	94.8	94.3	94.4	95.3	94.8
L31350	USA	--	1995	95.6	94.8	95.3	96.3	95.6
KU366270	Iran	Pumpkin	2013	95.8	95.4	95.5	96.3	95.8
MW449260	France	Zucchini	2022	96.2	96.3	96.0	97.3	96.2
MW449262	France	Melon	2022	94.3	94.1	94.0	94.5	94.3
MW449263	France	Melon	2022	93.8	93.6	93.1	94.6	93.8
MK033873	China	Pumpkin	2017	97.3	97.5	96.5	97.3	97.3
AY995216	New Zealand	Zucchini	2016	96.2	96.5	95.8	96.5	96.2
OQ335839	Italy	Pumpkin	2023	93.2	92.9	92.9	92.9	93.2
KJ614229	India	Bitter Gourd	2013	96.1	95.8	95.6	96.9	96.1
JF797206	India	Cucumber	2011	78.5	78.5	77.7	79.0	78.5
MW345250	Turkey	Cucumber	2009	93.7	93.7	93.3	94.4	93.7
OK558793	Canada	-	2001	93.9	93.9	93.3	94.6	93.9
AY188994	Israel	-	2004	77.6	77.6	77.1	78.3	77.6
D00692	USA	Zucchini	1990	93.7	93.9	93.1	94.6	93.7
OM471983	UK	Pumpkin	2022	95.9	95.3	95.6	96.5	95.9
LC726785	China	Vegetables	2019	95.7	95.0	95.2	96.2	95.7

**Table 4 plants-12-03503-t004:** Recombination events detected in Pakistan isolates of ZYMV sequences.

Recombinant Isolate	Breaking Point		Major Parent	Minor Parent	*p*-Value						
	Starting	Ending			RDP	GENECONV	BootScan	MaxChi	Chimera	SiScan	3SEQ
MK848239	266	814	MK848237	MK956829	NS	1.257 × 10^−2^	NS	3.121 × 10^−3^	3.925 × 10^−2^	2.045 × 10^−8^	5.253 × 10^−4^
KR261952	74	650	MN598576	MK848239	NS	NS	NS	1.295 × 10^−1^	2.269 × 10^−3^	1.143 × 10^−2^	9.417 × 10^−4^

## Data Availability

All the information and data have been added to the manuscript. However, the details of genetic sequences have been uploaded to GenBank, NCBI, and can be accessed using the accession numbers which are publicly available.

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
