# Peer review of "Zucchini Yellow Mosaic Virus (ZYMV) as a Serious Biotic Stress to Cucurbits: Prevalence, Diversity, and Its Implications for Crop Sustainability"

_plants, 2023, doi:10.3390/plants12193503_

Round 1

Reviewer 1 Report

Dear Colleagues.

There are a few notes

Author Response

The authors are thankful to the reviewers for their valuable input. The authors strongly believe the manuscript will significantly improve after making the suggested changes. Point-wise changes have been made and mentioned in the file attached.

Reviewer 2 Report

1)     The title should be changed and must reflect the aim of the work. The author mentioned that the study was conducted to estimate the incidence and molecular characterization of potyviruses, especially ZYMV, in major cucurbit crops in the Pothwar region.

2)     Table 1 missed the number of samples of each host; please add

3)     Figures 1 and 2 transfer to materials and methods section

4)     Line 407: A total of 31 resembling ZYMV CP gene sequences from NCBI database were downloaded and aligned with six isolates that are newly identified from Pakistan along with an isolate of CMV (outgroup) using CLUSTAL W program embedded in MEGA7 software.  Six or five isolates? please correct

5)     One of the major issues is that the authors mentioned that the coat protein length of ZYMV is 1110 nt encoding 369 aa of CP, although all previous literature  assessed that the CP is about 800 not; please check it again and revise it throughout the manuscript

6)     Line 143-145: Virtual gel resulting from RFLP simulation revealed that all the ZYMV isolates contain a distinct 43 bp band while the CMV outgroup isolates lacked it. The RFLP figure shows three  ZYMV isolates missed 43 bp band. Please check and correct

language: need to improve 

Author Response

(The authors gave the same response as above.)
